# Diagnosis, Progress, and Treatment Update of Kawasaki Disease

**DOI:** 10.3390/ijms241813948

**Published:** 2023-09-11

**Authors:** Ho-Chang Kuo

**Affiliations:** 1Kawasaki Disease Center, Kaohsiung Chang Gung Memorial Hospital, Kaohsiung 83301, Taiwan; erickuo48@yahoo.com.tw; 2Department of Respiratory Therapy, Kaohsiung Chang Gung Memorial Hospital, Kaohsiung 83301, Taiwan; 3Department of Pediatrics, Kaohsiung Chang Gung Memorial Hospital, Kaohsiung 83301, Taiwan; 4College of Medicine, Chang Gung University, Taoyuan 33302, Taiwan; 5Taiwan Association for the Promotion of Molecular Hydrogen, Kaohsiung 83301, Taiwan

**Keywords:** Kawasaki disease, diagnosis, biomarkers, treatment, eosinophil

## Abstract

Kawasaki disease (KD) is an acute inflammatory disorder that primarily affects children and can lead to coronary artery lesions (CAL) if not diagnosed and treated promptly. The original clinical criteria for diagnosing KD were reported by Dr. Tomisaku Kawasaki in 1967 and have been used for decades. However, research since then has highlighted the limitations of relying solely on these criteria, as they might lead to underdiagnosis or delayed diagnosis, potentially increasing the risk of coronary artery complications. This review appears to discuss several important aspects related to KD diagnosis and management. The current diagnostic methods for KD might need updates, especially considering cases that do not fit the typical clinical criteria. Recognizing diagnostic pitfalls and distinguishing KD from other conditions that might have similar clinical presentations is essential. The differences and similarities between KD and Multisystem Inflammatory Syndrome in Children (MIS-C), another inflammatory condition that has been associated with COVID-19, were also reviewed. The review explores the potential role of eosinophil count, new biomarkers, microRNA panels, and scoring systems in aiding the diagnosis of KD. Overall, the review article provides a comprehensive overview of the evolving landscape of KD diagnosis and management, incorporating new diagnostic methods, biomarkers, and treatment approaches to improve patient outcomes and reduce the risk of complications.

## 1. Introduction

Kawasaki disease (KD) was first described by Japanese pediatrician Dr. Tomisaku Kawasaki in 1967 in a Japanese-language journal and later in 1974 in an English-language journal. He observed a 4-year-old boy who exhibited a range of clinical symptoms, including a persistent high-grade fever and a skin rash. Initially, he referred to this condition as “acute febrile mucocutaneous lymph node syndrome” (MCLS). Although the new diagnosis faced initial skepticism, Dr. Kawasaki persisted. After collecting a series of 50 cases, he published his findings, accompanied by meticulous hand-drawn diagrams, in a Japanese medical journal. Dr. Kawasaki outlined the key features of this newly discovered disease, which included persistent fever, bilateral non-purulent conjunctivitis, diffuse oral fissures, a distinctive skin rash, edema of the hands and feet, as well as lymphadenopathy of the neck [1]. The diagnosis and treatment of KD need updating, including criteria comparison, the importance of eosinophils, novel markers and techniques for diagnosis, the potential role of hydrogen gas inhalation in coronary artery lesions (CAL), and precise treatment to prevent CAL formation.

## 2. Clinical Features and Diagnosis Criteria Comparison of Kawasaki Disease

Even though the initial name for this novel inflammatory disease was eventually altered to “Kawasaki disease” as a tribute to him, his original depiction of the condition still closely aligns with the current diagnostic criteria. Apart from Japan, the most widely adopted diagnostic criteria for KD are those established by the American Heart Association (AHA). These criteria encompass a fever lasting for at least 5 days (above 38 °C) and the presence of at least 4 out of the following 5 major clinical criteria [2]: bilateral conjunctival injection without discharge, changes in the oral mucosa (including fissured lips, strawberry tongue, or a red pharynx), alterations in the peripheral extremities (such as redness of the palms and soles, acute-phase edema of the hands and feet, and peeling of the skin around the nails after the acute phase), a diverse rash, and cervical lymphadenopathy (always with a size exceeding 1.5 cm in diameter) [2].In Japan, the diagnostic criteria for KD established by the Japanese Ministry of Health are primarily employed, and they exhibit minor variations. Notably, in contrast to the AHA criteria, the presence of fever for at least 5 days is not deemed an obligatory criterion under the Japanese guidelines. Put differently, individuals who display all five cardinal symptoms but are either without fever or have a fever lasting fewer than 5 days (1, 2, 3, or 4 days) can still receive a KD diagnosis based on the Japanese criteria [3].The comparison of different diagnostic criteria is presented in Table 1. Based on these diagnostic standards, we have developed the “Kuo’s 1-2-3-4-5 mnemonic,” a convenient memory aid for the five major KD criteria, as demonstrated in Table 2. In a 2017 update to the diagnostic guidelines for KD, the AHA acknowledged the ongoing discussion regarding fever duration. The AHA mentioned that KD can now be diagnosed in patients with a fever lasting a minimum of 4 days (as opposed to the previous requirement of 5 days), provided they also exhibit at least four of the five cardinal symptoms. This is particularly applicable if there are signs of redness or swelling of the palms or soles or if there is edema in the hands and feet [2].We term this the “4-4-4 rules” (coined by Prof. HC Kuo), encompassing the requirement for a minimum of 4 days of fever, the presence of more than 4 major criteria, and changes in 4 limbs. These diagnostic criteria have gained widespread acceptance in both research and clinical contexts. It is essential to recognize that the cardinal manifestations of KD might not all manifest concurrently and might even diminish before an accurate diagnosis can be established. Consequently, a comprehensive medical history and repeated assessments (conducted every other day) are crucial for the precise and prompt identification of KD.

Additional minor clinical symptoms that could be evident in KD patients but are not encompassed by the major diagnostic criteria consist of arthritis, gastrointestinal involvement (with a lower incidence compared to multisystem inflammatory syndrome in children, MIS-C), irritability, lethargy, neurological manifestations, cough, and rhinorrhea (with a lower incidence compared to MIS-C). Arthritis, primarily affecting the large joints of the lower extremities (such as the knees, hips, and ankles), can be identified in 7.5% to 25% of patients and is typically temporary and non-deforming [4].Arthritis following the acute phase of KD also suggests ongoing inflammation, prompting the consideration of anti-inflammatory treatment involving steroids for these children. In a study involving 198 KD patients, additional prodromal symptoms experienced during the acute phase of the disease encompassed irritability in 50% of patients, vomiting in 44%, diarrhea in 26%, and abdominal pain in 18% [5]. In exceptional instances, abdominal imaging techniques such as radiographs or computed tomography (CT) scans might reveal indications of pseudo-obstruction, a condition that can sometimes manifest prior to the appearance of cardinal symptoms [6]. KD should also be taken into consideration in the differential diagnosis of infants and children who exhibit a prolonged fever lasting over 7 days and unexplained aseptic meningitis. Among the 1582 KD patients reviewed, 80 (5.1%) displayed evidence of neurological engagement. Lethargy was the predominant symptom (50.1%), followed by irritability (26.3%), meningeal irritation (18.8%), convulsions (17.5%), headache (16.3%), bulging fontanelles (8.8%), and facial palsy (1.3%) [7]. Based on the summary derived from our compilation of 110 KD cases, the prevalent characteristics of the disease are represented by rashes, desquamation, conjunctivitis, and strawberry tongue (occurring in over 90% of cases); induration (75–80%); pyuria (50%); BCG induration (35–40%); neck lymphadenopathy (25–30%); diarrhea (21–25%); and arthritis (5–10%) [8]. The figure from our previous publication was included in the Nelson Textbook of Pediatrics, 20th edition, page 1210. The clinical symptom spectrum of KD is shown in Figure 1 below.

Yet another clinical manifestation of KD is Kawasaki disease shock syndrome (KDSS), which, while uncommon, is severe. It is characterized by shock, hypotension, compromised left ventricular systolic function, mitral regurgitation, consumptive coagulopathy, and a notable heightened risk of both CAL and a lack of response to intravenous immunoglobulin (IVIG) treatment [9]. Due to clinical resemblances, including symptoms such as rash, fever, and shock, patients afflicted with KDSS might be erroneously diagnosed as having toxic shock syndrome [10]. KD may also manifest as cervical lymphadenitis and exhibit contiguous cellulitis and phlegmon that mimic bacterial adenitis [11], especially in children less than 6 months old or more than 4–5 years old [12]. The potential for KD should be taken into account when evaluating infants and children (over 4–5 years old) who present with extended fever, cervical lymphadenitis, and retropharyngeal or parapharyngeal phlegmon, especially if they show poor response to antibiotic treatment [2].

In countries such as Taiwan, China, or Japan, where vaccination programs routinely administer the Bacillus Calmette-Guérin (BCG) vaccine, around 50% of KD patients may exhibit induration, erythema, or crusting around the BCG vaccination site. These symptoms can potentially assist in the diagnosis of KD, particularly among children under 3 years of age [8,13]. BCG site induration is a distinctive clinical symptom of KD, although it is not included in the five major criteria of either the AHA or Japanese guidelines. In a previous study involving 34 KD patients, the induration around the BCG vaccination site was categorized into three prevalent patterns: a targetoid bull’s eye pattern centered around the BCG site, uniform erythema surrounding the BCG site, or a whitish patch at the BCG inoculation site. Patients displaying a targetoid bull’s eye pattern around the BCG site during KD diagnosis were correlated with an elevated risk of coronary artery dilatation [14].For patients who have received the BCG vaccine and meet only four of the primary symptoms outlined in the Japanese guidelines, the presence of erythema, induration, or crusting around the BCG vaccination site could offer a significant indicator highly suggestive of a KD diagnosis. Erythema at the BCG vaccination site is relatively specific to a diagnosis of KD, although it has also been observed in connection with measles and human herpesvirus type 6 (HHV6) infections [15]. The existence of BCG infections in febrile children serves as a strong reason for clinicians to consider the potential of KD. The BCG vaccination reaction in KD is thought to involve a T-cell immune response. Uehara et al. reported and proposed that redness at the BCG vaccination site is a valuable indicator for diagnosing KD, particularly in nations with a BCG vaccination regimen. Even when patients display 4 or fewer of the 5 major clinical criteria for KD, doctors should be aware that individuals with BCG site reactions could potentially be dealing with KD [13].

## 3. The Importance of Induration over Peripheral Extremities in Kawasaki Disease

Our research team devised a wireless optical monitoring system that employed a tri-wavelength light source (700, 910, and 950 nm) along with a light detector to assess the extent of edema in the palm and sole tissues of febrile patients under suspicion for KD. In comparison to age-matched febrile controls, KD patients exhibited notably higher relative peripheral edema with increased water concentrations and decreased total hemoglobin concentrations. Subsequently, we documented alterations in tissue hemoglobin and water levels at varying stages of KD through near-infrared spectroscopy detection. Our findings demonstrated a significant correlation between water content and the development of CAL [16]. As a result, this non-invasive device could offer valuable assistance to physicians in promptly identifying KD and differentiating it from other fever-related conditions. We hold a positive outlook that wireless optical monitoring has the potential to furnish an extra non-invasive diagnostic tool, particularly for patients suspected of having KD but not meeting the conventional clinical criteria [17]. The presence of limb indurations and the elevation of water content detected through near-infrared spectroscopy over the palm region align well with the 2017 AHA 4-4-4 rule. This underscores the significance of indurations in the peripheral extremities as a diagnostic indicator for KD.

## 4. Diagnosis of Incomplete (or Atypical) Kawasaki Disease

Up to 10–15% of KD patients whose diagnosis is confirmed through an echocardiogram do not fulfill the criteria mentioned earlier. This subgroup of patients is often referred to as having “incomplete” or “atypical” KD [18], especially in infants younger than one year old [19,20] and children older than 5 years old. In 2004, the AHA introduced guidelines to assist in diagnosing incomplete KD, offering supplementary criteria. For patients with incomplete KD, the administration of high-dose IVIG (2 gm/kg) is recommended for those who meet all four of the following criteria: (1) fever lasting 5 days or more; (2) presence of two or three clinical KD symptoms; (3) elevated C-reactive protein (CRP) level exceeding 3.0 mg/dL or an erythrocyte sedimentation rate (ESR) surpassing 40 mm/h; and (4) fulfillment of at least three of the supplementary laboratory criteria (age-appropriate anemia, platelet count of ≥450,000/mm^3^ after the 7th day of fever, albumin level ≤3.0 g/dL, elevated alanine transaminase, white blood cell count of ≥15,000/mm^3^, urine white blood cells ≥10/high-powered field) or a positive echocardiogram (Table 3) [21]. Additionally, for patients exhibiting a fever lasting 5 or more days and two or three of the cardinal symptoms of KD, it is advisable to conduct sequential clinical and laboratory monitoring every other day in case the fever persists. Moreover, if there is periungual peeling of the fingers and toes, it is recommended to undergo supplementary echocardiography.

Diagnosing KD can be more challenging in infants and older children above 5 years of age. Infants under 6 months old are more likely to exhibit incomplete KD, which can result in delayed IVIG therapy and an increased risk of coronary artery involvement. This risk persists even among those who receive IVIG therapy within 10 days of the disease’s onset [22,23]. In an analysis of 113 KD patients, those who were older than 5 years old exhibited cardinal KD symptoms later in the clinical progression. They were also inclined to display more pronounced signs of inflammation, characterized by elevated levels of erythrocyte sedimentation rate (ESR) and a prolonged duration of fever. This group was also at a heightened risk of being resistant to IVIG treatment. Intriguingly, older children were also more likely to experience cervical lymphadenopathy, and in some cases, their initial presentation occurred after the onset of fever. This pattern was in contrast to children under 5 years old, among whom the initial presentation often took place alongside fever (85.0% vs. 51.6%) [24].

Numerous viral infections, such as adenovirus, measles, and Epstein-Barr virus (EBV), exhibit clinical symptoms that resemble those of KD, such as fever, rash, and inflammation of the mucous membranes of the eyes and mouth [2]. Recognizing that KD can coincide with both bacterial and viral infections is essential, and it is important to note that confirming a bacterial or viral infection does not necessarily rule out the possibility of a KD diagnosis [25,26]. Lee et al. reported that the likelihood of IVIG resistance rises with higher CRP values and the utilization of multiple intravenous (IV) antibiotics. This suggests that physicians should exercise caution when administering multiple IV antibiotics to treat suspected infections in children with KD [27]. Administering antibiotics is unnecessary and not recommended as a standard treatment when diagnosing KD in the absence of evidence indicating a bacterial infection.

## 5. Echocardiographic Findings Aid to Make Precise Diagnosis of Kawasaki Disease

Echocardiography should be conducted for all individuals suspected of or confirmed to have KD. This practice is valuable not only for aiding KD diagnosis but also for establishing a baseline of echocardiographic parameters for future monitoring. The assessment should cover the visualization of all major segments of the coronary artery, including observations of diameter dilatation, irregular arterial wall, non-tapering, or increased echogenicity of the artery wall. Additionally, ventricular wall motion, ejection fraction, valvular regurgitation, fistula formation, and the presence of pericardial effusion should be monitored. It is strongly recommended to use a high-frequency transducer in both infants and older children to enhance coronary artery visualization [2]. CAL can reach their maximum diameter within four to six weeks after the onset of KD and may subsequently regress over a span of one to two years or even longer, depending on the size of the aneurysm. Consequently, having a normal echocardiogram during the initial week of disease onset does not rule out a diagnosis of KD. In cases where there is a strong suspicion of KD, it is advisable to undergo serial examinations every other day.

Measuring the internal diameter of coronary arteries is critical and can be categorized using absolute dimensions, as outlined in the Japanese JCS criteria [3,28] or be adjusted according to the patient’s body weight and body height of body surface area (AHA Guidelines) [2]. According to the Japanese guidelines, coronary arteries are considered dilated if the internal diameter exceeds 3 mm in a child younger than 5 years old or 4 mm in a child older than 5 years old. Coronary artery dilatations can be categorized as follows: (Figure 2).

Small aneurysm: 3 mm to 4 mm in children younger than 5 years old; >4 mm or if the internal diameter is less than 1.5 times that of an adjacent segment in children older than 5 years old.

Medium aneurysm: Internal diameter of 4 mm to 8 mm in children younger than 5 years old; 1.5 to 4 times the size of an adjacent segment in children older than 5 years old.

Giant aneurysm: Internal diameter exceeding 8 mm in children younger than 5 years old, or if it measures more than 4 times that of an adjacent segment in children older than 5 years old.

However, grading coronary artery size based solely on absolute measurements of diameter and age does not consider differences in body size or height, which can result in underestimating the extent of coronary artery involvement in up to 27% of patients [29]. Hence, significant endeavors have been directed towards standardizing coronary artery dimensions across various age and weight groups through body surface area (BSA) adjustments. The Z-score, which indicates the number of standard deviations from the mean values, has been adopted as the foundation for evaluating coronary artery diameters in line with the AHA guidelines. In this context,

Coronary artery diameters are considered normal if Z-scores are less than 2.

Dilation is indicated if Z-scores range from 2 to 2.5.

Small aneurysms are characterized by Z-scores of 2.5 to 5.

Medium aneurysms encompass a Z-score of 5 to 10, along with an absolute measurement of less than 8 mm.

Large or giant aneurysms correspond to a Z-score exceeding 10 or an absolute measurement surpassing 8 mm. Indeed, the Z-score offers a more accurate assessment of coronary artery dilation compared to the traditional method of measuring coronary artery diameter.

## 6. Novel Biomarkers to Assist in the Diagnosis of Kawasaki Disease

Numerous inflammatory markers, including erythrocyte sedimentation rate (ESR), C-reactive protein (CRP) values, white blood cell (WBC) counts, and platelet counts, can contribute to the diagnosis of KD, either in combination with clinical symptoms or independently. However, these inflammatory markers are generally nonspecific to KD and can also become elevated in other conditions involving infection, inflammation, or autoimmunity. For instance, in a study involving 114 patients under suspicion of KD, an ESR level of ≥40 mm/h exhibited a sensitivity of 90.5% but a specificity of merely 66.6% [30]. Researchers at Stanford University have formulated two scoring systems aimed at distinguishing between KD and febrile controls. The initial scoring system, introduced in 2013, employed five clinical symptoms and 12 laboratory data points (a total of 17 parameters) to categorize febrile patients into low-risk (with a negative predictive value exceeding 95%), intermediate-risk, and high-risk (with a positive predictive value greater than 95%) KD groups [31]. Nevertheless, even after this stratification, a portion of febrile patients (approximately 20–30%) could not be classified. In response, a subsequent algorithm involving 18 clinical and laboratory data points were devised. This new algorithm employed data-mining models to re-stratify patients with the aim of enhancing the identification of KD among children presenting with fever [32] which was further confirmed by a Taiwanese cohort with 418 KD and 259 FC from Kaohsiung Chang Gung Memorial Hospital [33].

We endeavored to create a more streamlined scoring system that effectively distinguishes between patients with fever and those with KD. After analyzing 6310 febrile patients and 485 KD patients, we constructed a scoring system employing solely eight laboratory criteria. Notably, the highest scores were assigned to eosinophil percentage exceeding 1.5%, CRP exceeding 24 mg/L, and alanine aminotransferase level exceeding 30 U/L [34]. The present clinical symptoms of KD did not enroll in this algorithm to predict the possibility of KD.

Prior research has indicated that KD is linked to heightened expression of various T helper (Th)-1 cytokines, such as IL-6, IL-12, TNF-alpha, CXCL10 (also known as IFN-γ-inducible protein 10 [IP-10]), and IFN-gamma. Additionally, elevated levels of Th2 cytokines, including IL-4, IL-5, IL-13, and IL-31, have been associated with KD [35,36,37]. Both Th1 and Th2 immune responses exhibit elevation during the acute stage of KD. Notably, the Th2 immune response appears to exert certain anti-inflammatory effects.

B-type natriuretic peptide (BNP) and its inactive cleavage product, N-terminal prohormone of brain natriuretic peptide (NT-proBNP), are investigated as proteomic biomarkers for KD. BNP is generated by ventricular cardiomyocytes as a response to ventricular stretching. It is recognized as a well-established biomarker for both congestive heart failure and coronary artery diseases [38]. The precision of NT-proBNP as a diagnostic biomarker for distinguishing KD from other febrile illnesses was recently assessed in a meta-analysis encompassing six studies involving a total of 279 KD patients. In this analysis, the biomarker demonstrated a combined sensitivity of 89% and a combined specificity of 72% [39]. Age-specific variations in NT-proBNP levels have been observed, with the highest levels occurring in the initial days after birth. These levels subsequently undergo a rapid decline during the first few weeks of life and continue to gradually decrease with age. These fluctuations make it challenging to establish a definitive cut-off value for NT-proBNP [40].

*Escherichia coli* (*E. coli*) proteome microarrays comprise approximately 4200 purified *E. coli* proteins and are utilized to examine the presence of anti-*E. coli* protein IgG and IgM antibodies in the patient’s serum samples. [41]. Contemporary theories of KD propose that the condition arises from an exaggerated immune response triggered by a common environmental or infectious factor. However, a universally acknowledged infectious trigger for KD has not yet been definitively identified [42]. Case reports have presented instances suggesting that *E. coli* and other Gram-negative pathogens, including Klebsiella oxytoca, could potentially serve as infectious triggers for KD [43,44]. Research focused on the gastrointestinal microbiota of KD patients has revealedincreased quantities of both Gram-positive and Gram-negative bacteria that produce heat shock proteins in the stool samples of individuals with KD [45]. *E. coli* is a prevalent bacterium in the gut and plays a role in the establishment of immune balance and the potential emergence of autoimmunity in young children [46]. Collectively, these investigations propose that *E. coli* might be a pertinent pathogen linked to the progression of KD. Pathogen-associated molecular patterns (PAMPs) and Toll-like receptors may also play a role in the context of KD. Our research revealed that individuals with KD exhibit distinct antibody profiles against *E. coli*, underscoring the significant role that *E. coli* potentially holds in the development of KD [47]. *E. coli* proteome microarrays have been previously documented in various diseases, including inflammatory bowel disease [48] and bipolar disorder [49]. Given that microarray testing using *E. coli* proteome microarrays necessitates only 125 picoliters of serum (equivalent to less than a single drop of blood), this method could potentially serve as an innovative approach for screening and aiding in the diagnosis of KD.

MicroRNAs (miRNAs) are short, non-coding RNA molecules comprising approximately 22 nucleotides. These molecules play a role in regulating gene expression by impeding the translation of mRNA into proteins. MiRNAs are present in various cell types, including erythrocytes, leukocytes, and platelets. Recent research indicates that miRNAs enclosed within exosomes can be detected in plasma, potentially engaging in gene regulation and mediating communication between distant cells [50]. A significant portion of the microRNAs detected in the serum of KD patients could potentially influence the growth and functionality of vascular endothelial cells. Among these microRNAs, miR-233 stands out as one of the most prominently expressed in the serum of individuals with KD [51]. Genes such as IGF1R and IL-6ST are among the targets for miR-233 [52]. MiR-233-3p directly targets the 3′ untranslated regions (UTR) of IL-6ST and effectively inhibits the expression of the crucial inflammatory cytokine IL-6 in KD. This inhibition consequently leads to a reduction in the expression of pivotal transcription factors such as p-STAT3 and NF-kB p65. In a KD mouse model, administration of miR-233-3p demonstrates a mitigation of vascular endothelial damage and the suppression of expression in key vascular adhesion molecules, including E-selectin and ICAM-1, as well as IL-6 [53].

Several other microRNAs have been documented to trigger apoptosis, such as miR-186 and miR-125-5p, or impede vascular cell proliferation, as observed with miR-27b. Notably, serum miR-186 has been identified as a promoter of endothelial cell apoptosis through the involvement of the SMAD6 and MAPK pathways [54]. Similarly, miR-125-5p has been observed to initiate endothelial cell apoptosis by engaging the MKK7 and Caspase-3 pathways [55]. The elevation of miR-27b in an endothelial cell line leads to the inhibition of the TGF-beta pathway and subsequent suppression of endothelial cell proliferation and migration, facilitated by SMAD7 [56]. Serum obtained from KD patients has exhibited decreased levels of miR-483. This microRNA is known to target the untranslated region of connective tissue growth factor (CTGF), a factor implicated in coronary artery remodeling and fibrosis. Suppressed miR-483 expression in endothelial cells has been associated with heightened CTGF expression [57].

MicroRNAs (miRNAs) have also been linked to prognostic outcomes in KD. In a study involving 102 KD patients and 18 healthy controls, it was observed that KD patients who were resistant to IVIG treatment displayed notably elevated levels of miR-200c and miR-371-5p [58]. The progression of CAL has also been correlated with the upregulation of certain microRNAs. These include miR-92a-3p, miR-let-7i-3p, miR-145-5p, and miR-320a [59,60]. The transfection of miR-145-5p and miR-320a into THP-1, a monocyte cell line, resulted in heightened expression of the inflammatory cytokines IL-6 and TNF-α. Moreover, immunohistochemical staining of a coronary artery sample obtained from KD patients exhibited escalated expression of miR-145-5p and miR-320a within the endothelial cells [61].

Identifying a single miRNA might not be as effective as determining a complete miRNA expression profile in distinguishing KD from fever controls. This is due to the likelihood that around 60% of protein-encoding genes are co-regulated by multiple miRNAs simultaneously [62]. In our research involving 50 KD patients, miRNAs extracted from peripheral leukocytes were subjected to next-generation sequencing (NGS), allowing us to identify a total of 10 miRNAs. This selection of 10 miRNAs was subsequently employed as a screening panel for diagnosing KD within the validation set. Notably, this panel demonstrated a sensitivity of 83.3% and a specificity of 92.5% [63] Furthermore, the expression levels of these miRNAs exhibited a strong correlation with the diagnostic indicators of KD. By amalgamating these miRNAs into a single KD miRNA signature utilizing the SVM (Support Vector Machine) model, the discriminatory ability was significantly enhanced, achieving a discrimination power of 0.882. This signature displayed a sensitivity of 74.6% and a specificity of 89.9% when evaluated across 665 cases (data not yet published). It is important to note that this miRNA panel has been patented in multiple regions, including China, Taiwan, Hong Kong, Japan, and the U.S., for the purpose of distinguishing KD from other febrile diseases.

## 7. Elevated of Eosinophil in KD

The higher expression of eosinophils, which can increase even more after IVIG treatment, has been associated with IVIG treatment responsiveness in Kawasaki disease (KD). This suggests a potential protective effect of eosinophils in KD. Interestingly, KD patients who exhibit lower Th2 immune responses (such as IL-5 or eosinophil levels) have been found to be at a higher risk of developing CAL. This implies that Th2 responses may play a protective role against the development of CAL in KD. This highlights the complexity of immune responses and their potential impact on the outcomes of KD [36]. Liu et al. [64] conducted a study involving 800 children, out of whom 249 were diagnosed with Kawasaki disease (KD) and 551 were age- and gender-matched children with non-KD febrile infectious diseases. In this study, both univariate and multivariate logistic regression analyses were employed, along with the development of nomogram models to analyze the experimental data. The children were divided into two groups:A total of 562 in the model group and 238 in the verification group.The prediction nomogram was constructed based on several factors, each assigned a certain point value: high eosinophil percentage (100 points), high C-reactive protein (CRP) levels (93 points), high alanine aminotransferase (ALT) levels (84 points), low albumin levels (79 points), and high white blood cell (WBC) count (64 points). The collective factors led to an area under the curve (AUC) of 0.873 for the model group and 0.905 for the validation group. Among all the parameters derived from routine laboratory data, eosinophils showed the highest odds ratio (OR) of 5.015 (95% CI: 3.068–8.197) during the multiple logistic regression analysis. In the validation group, the sensitivity was 84.1% and the specificity was 86%. This indicates that eosinophils play a crucial role in the nomogram model as the most significant predictor of KD.

Tsai et al. [34] conducted a comprehensive analysis involving 6310 febrile children and 485 subjects diagnosed with Kawasaki disease (KD). The study focused on evaluating routine blood test parameters, which included complete blood count with differential (CBC/DC), C-reactive protein (CRP), aspartate aminotransferase, and alanine aminotransferase. They utilized various statistical tools such as the receiver operating characteristic curve, Youden’s index, and logistic regression model to construct a prediction model. The research identified eight independent predictors that could differentiate between KD and other febrile illnesses. Among these predictors, eosinophils >1.5% had the highest score [7], followed by alanine aminotransferase >30 U/L [6] and CRP >25 mg/L [6]. A total score of 14 (from a maximum of 30) yielded the best prediction rate, combining sensitivity and specificity for KD. The calculated sensitivity, specificity, and accuracy values were 0.824, 0.839, and 0.838, respectively. Verification tests were carried out on two independent cohorts from different hospitals (Kaohsiung, Taiwan, and Shenzhen, China), comprising 273 subjects. The validation showed a sensitivity of 0.780 (213/273) for accurately identifying KD cases. Remarkably, this study exclusively utilized routine laboratory data from CBC/DC, CRP, and liver enzyme levels (GOT/GPT) without considering factors such as age, gender, clinical symptoms, or urine findings. The study highlighted the pivotal role of eosinophils in distinguishing KD from other febrile illnesses, suggesting the potential inclusion of eosinophils as an independent factor in the supplementary diagnostic criteria for KD, as outlined by the AHA.

## 8. Consulting a Clinical Expert on Kawasaki Disease

When dealing with children who exhibit prolonged fever exceeding 7 days without a clear diagnosis, seeking consultation from a team of Kawasaki disease (KD) specialists is essential. This specialized team would ideally include experts from various fields, such as immunology, rheumatology, cardiology, infectious diseases, and KD clinical expertise. Collaborating with a multidisciplinary team ensures accurate diagnosis and timely treatment to prevent the development of CAL. The diagnostic process for KD, which relies on five key clinical symptoms, can introduce subjectivity. Involving a clinical expert in KD can enhance the accuracy and objectivity of the diagnosis. Consulting experts who possess vast experience and knowledge in the field of KD increases the likelihood of an accurate assessment. To assist in identifying qualified KD experts, the expertscape directory provides a valuable resource. This directory offers objective rankings of medical expertise and can be searched based on various criteria such as city, region, country, and continent. It is important to prioritize clinical experts who specialize in KD over research experts, as clinical expertise is particularly relevant for accurate diagnosis and patient care.

## 9. IVIG Resistance (IVIG Non-Responsiveness or IVIG Failure) in Kawasaki Disease

KD patients with IVIG resistance have a higher risk of developing CAL. Identifying high-risk patients who may benefit from more aggressive treatment is important. Administering a second dose of IVIG (2 g/kg over 10–12 h, following the same dosage and treatment duration as the initial IVIG) is a consideration [65,66] methylprednisolone pulse therapy [67] tumor necrosis factor-alpha blockade [68] cyclophosphamide; cyclosporine A; plasmapheresis [69] methotrexate [70] and plasma exchange [71] have all been reported to be beneficial for KD patients with initial IVIG-resistance. IVIG is recognized as the primary standard treatment for KD in accordance with AHA and Japanese guidelines. However, there is currently no established standard treatment for KD cases resistant to IVIG. The utilization of single-pulse intravenous methylprednisolone (IVMP), with a dose of 30 mg/kg (up to a maximum of 1000 mg), in conjunction with initial IVIG has not demonstrated a significant improvement in the disease’s outcome for children with KD [72]. While the addition of IVMP to IVIG for initial treatment does not appear to enhance therapeutic efficacy, the underlying mechanism for this remains unclear. However, it is worth noting that administering IVMP therapy over a span of three days does seem to offer benefits for patients with IVIG-resistant KD [66]. Steroid receptor expression changes following IVIG treatment in KD could potentially shed light on the role of IVMP pulse therapy after IVIG, although not in combination with IVIG for KD. At Kaohsiung Chang Gung Memorial Hospital in Taiwan, we adopted a two-step approach for managing IVIG-resistant KD cases. Initially, a second course of high-dose IVIG (2 g/kg over 10–12h) was administered for patients who showed resistance to the initial IVIG treatment. For those who continued to be resistant to IVIG, a secondary dose of IVMP (30 mg/kg/day for 3 days) was prescribed. In instances of ongoing resistance even after IVMP, an anti-TNF-alpha agent such as infliximab (5 mg/kg) was utilized.

## 10. Potential Role of Molecular Hydrogen Gas (H_2_) in Kawasaki Disease

Oxidative stress, inflammation, and the generation of free radicals are all interconnected processes that play a significant role in the pathogenesis of KD and the formation of CAL [73]. Oxidative stress occurs when there is an imbalance between the production of reactive oxygen species (ROS) and the body’s ability to detoxify them through antioxidants [74]. In KD, the immune response and inflammation triggered by an infectious or environmental trigger can lead to the production of ROS and oxidative stress. This oxidative stress can damage cellular components, including lipids, proteins, and DNA. The endothelial cells lining the coronary arteries are particularly susceptible to oxidative damage, which can contribute to inflammation, vasculitis, and ultimately the development of CAL. Oxidative stress also promotes the release of pro-inflammatory cytokines and chemokines, amplifying the immune response and contributing to tissue damage. The resulting inflammation and tissue injury further exacerbate oxidative stress, creating a cycle that can lead to the progression of KD and the formation of CAL. It has been reported that molecular hydrogen gas (H_2_) exhibits beneficial effects in a variety of diseases by reducing oxidative stress. These diseases include rheumatoid arthritis (RA), atopic dermatitis (AD), hay fever, asthma, chronic obstructive pulmonary disease (COPD), COVID-19, severe COVID-19, stroke, depression, dementia, post-cardiac arrest syndrome, subarachnoid hemorrhage, myocardial infarction, chronic kidney disease, sepsis, hemorrhagic shock, and various cancers [75,76]. Hydrogen gas inhalation was reported to have a role in the treatment of COVID-19, reducing disease progression by improving airway resistance and inflammation [76]. Furthermore, a preliminary investigation has reported that inhaling hydrogen gas can lead to a significant improvement in breathing difficulties for the majority of COVID-19 patients. [77]. We conducted the initial study on a KD-related aneurysm (measuring 6.08 mm in diameter and 35 mm in length) that exhibited regression (complete regression at the 4-month follow-up, on day 138 of the illness) with the addition of supplementary therapy involving hydrogen gas inhalation, and no other complications were observed. Hydrogen gas inhalation could potentially serve as an alternative therapy for KD by acting as an antioxidant or anti-free radical agent, although further research is still necessary [78]. Cardiovascular diseases such as post-cardiac arrest syndrome, subarachnoid hemorrhage, and myocardial infarction are also associated with oxidative stress and inflammation. Hydrogen’s antioxidant and anti-inflammatory effects might contribute to cardiovascular protection and the possibility of treatment effects for KD, MIS-C, or COVID-19;however, we still need more clinical studies. The potential therapeutic impact of hydrogen gas inhalation in patients with KD, whether with or without CAL formation, also demands further investigation. Additionally, COVID-19 has been associated with the emergence of a KD-like condition known as multisystem inflammatory syndrome in children (MIS-C), representing a novel syndrome linked to SARS-CoV-2 infection in pediatric populations [79,80,81]. The comparison between KD and multisystem inflammatory syndrome in children (MIS-C) is shown in Table 4.

## 11. Conclusions

The Kuo mnemonic, 4-4-4 rule, BCG vaccination induration, AHA supplementary criteria, and strategic echocardiography scheduling collectively contribute to the prompt diagnosis of KD and the prevention of CAL formation. Biomarkers such as eosinophil count, miRNA expression, inflammatory cytokines, and *E. coli* antibody profiles have demonstrated significance in differentiating KD from other febrile conditions. Seeking advice from KD specialists, appropriate and intensified interventions for IVIG resistance, and adjunctive therapies such as hydrogen gas inhalation can further enhance the therapeutic outcomes for KD.

## Figures and Tables

**Figure 1 ijms-24-13948-f001:**
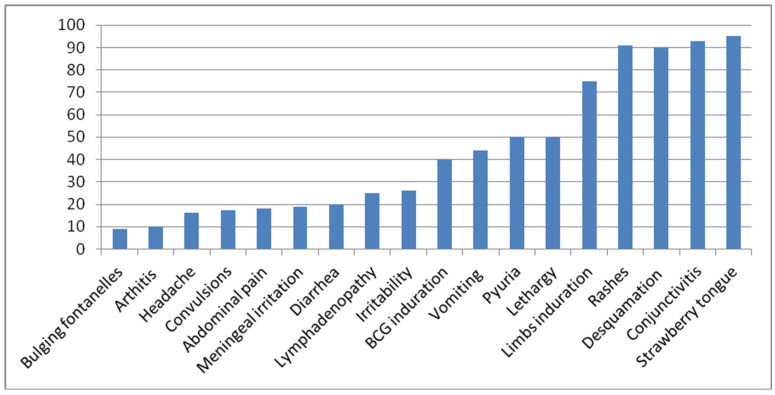
Percentage of clinical symptoms and signs of Kawasaki disease.

**Figure 2 ijms-24-13948-f002:**
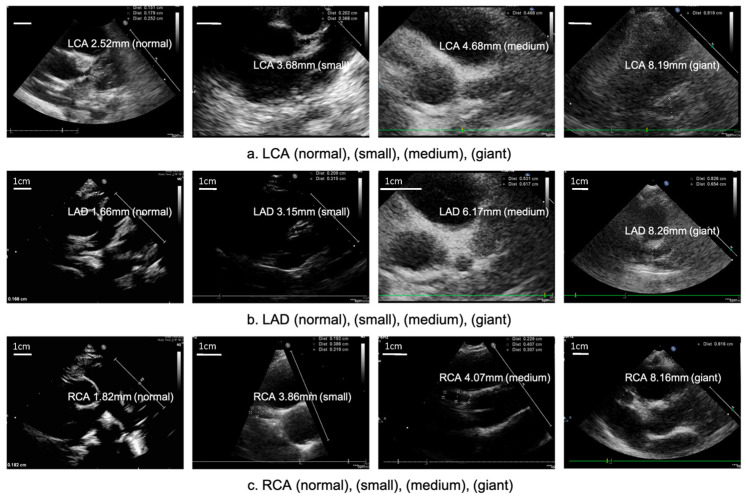
Coronary artery dilatations can be categorized as normal, small, medium and giant size (left to right). LCA: left coronary artery (**a**); LAD: left anterior descending artery (**b**); RCA: right coronary artery (**c**). (left upper white bar means 1 cm in scale in each figures).

**Table 1 ijms-24-13948-t001:** Updated Diagnostic Criteria for Kawasaki Disease.

Kawasaki Disease Research Committee (Japanese Ministry of Health) [3]	American Heart Association Guidelines [2]
Principal symptoms (at least five of the following): 1. Fever for more than 5 days 2. Bilateral conjunctivitis 3. Oral mucosal changes (diffuse injection, strawberry tongue, or fissured lips) 4. Skin rash 5. Changes in peripheral extremities (initial stage: Redness and swelling of the palms and soles; convalescent stage: peeling of the fingertips) 6. Acute non-purulent cervical lymphadenopathy (more than 1.5 cm in diameter) * Patients with four of the principal symptoms can be diagnosed with KD if a coronary aneurysm or dilatation is detected by 2D echocardiography or coronary angiography. ** According to this Japanese criteria, KD can be diagnosed on the 1st day, 2nd day, 3rd day, or 4th day of a fever that does not fit the AHA guidelines.	Fever for at least 5 daysand at least four of the following five criteria: 1. Erythema and cracking of lips, strawberry tongue, and/or erythema of oral and pharyngeal mucosa 2. Bilateral bulbar conjunctival injection without exudate 3. Rash: maculopapular, diffuse erythroderma, or erythema multiforme-like 4. Erythema and edema of the hands and feet in the acute phase and/or periungual desquamation in the subacute phase 5. Cervical lymphadenopathy (≥1.5 cm diameter), usually unilateral.
**4-4-4 Rules by AHA**	**Incomplete (Atypical) Kawasaki Disease-(AHA Supplementary Criteria)**
In the 2017 update of diagnostic guidelines for KD, the AHA acknowledged the debate regarding fever duration and stated that KD can still be diagnosed in patients with a fever lasting more than 4 days who also present with at least four of the five major clinical symptoms, especially if palmar or plantar erythema or edema of the hands and feet (4 limb changes) are present (we call this the 4-4-4 rule).	Fulfill all four of the following criteria: 1. Fever of 5 days or more 2. At least two major clinical symptoms of KD 3. CRP ≥ 3.0 mg/dL OR; ESR ≥ 40 mm/h 4. At least three of the following six supplementary laboratory criteria: anemia for age, platelet count of ≥450,000/mm^3^ after the 7th day of fever, albumin ≤3.0 g/dL, elevated alanine transaminase, white blood cell count of ≥15,000/mm^3^, urine white blood cells ≥10/high powered field.

* and ** author’s comment for the criteria comparison.

**Table 2 ijms-24-13948-t002:** Rapid memory of Kuo’s 1-2-3-4-5 mnemonic for Kawasaki disease.

Number	Mnemonic	Clinical Features
1	One mouth	Strawberry tongue, fissured lips, injected pharynx, and other signs of oropharyngeal mucosa inflammation
2	Two eyes	Bilateral conjunctivitis(without discharge)
3	Three fingers to check cervical lymphadenopathy	Cervical lymph node enlargement of >1.5 cm in diameter of at least one lymph node
4	Four extremities changes	Erythema of the palms and soles, edema of the hands and feet in the acute phase, or periungual desquamation after the acute phase
5	Five days of fever and polymorphous rash	Fever for more than 5 days; much skin rash

**Table 3 ijms-24-13948-t003:** Echocardiography Findings in Kawasaki Disease.

American Heart Association (AHA) Guidelines: [2]	Japan Circulation Society Guideline: Classification of Coronary Aneurysms during the Acute Phase
Positive findings for KD Include: 1. Left anterior descending coronary artery or right coronary artery with a Z-score ≥ 2.5 2. Coronary artery aneurysm formation 3. ≥3 of the following suggestive features: mitral regurgitation, pericardial effusion, decreased left ventricular function, or Z-scores in the left anterior descending coronary artery or right coronary artery of 2 to 2.5. Z-score classification of coronary artery lesions by AHA: 1. Dilatation only: Z-score 2 to 2.5 2. Small aneurysm: Z-score ≥ 2.5 to 5 3. Medium aneurysm: Z-score ≥ 5 to 10 or absolute measurement <8 mm 4. Large aneurysm: Z-score ≥ 10 or absolute measurement ≥8 mm	1. Small aneurysms or dilatation: Children < 5 years old: localized dilatation of coronary artery >3 mm but within ≤4 mm internal diameter Children ≥ 5 years old: localized dilatation of coronary artery >4 mm or if the internal diameter of a segment measures <1.5 times that of an adjacent segment 2. Medium aneurysms: Children < 5 years of age: an internal diameter of the coronary artery from >4 mm to <8 mm Children ≥ 5 years of age: the internal diameter of a coronary artery segment measures 1.5–4 times that of an adjacent segment 3. Giant aneurysms: Children < 5 years of age: coronary artery with an internal diameter of ≥8 mm Children ≥ 5 years of age: the internal diameter of a coronary artery segment measures >4 times that of an adjacent segment

**Table 4 ijms-24-13948-t004:** Comparison between Kawasaki disease (KD) and multisystem inflammatory syndrome in children (MIS-C).

	Kawasaki Disease (KD)	Multisystem Inflammatory Syndrome in Children (MIS-C)
Etiology	unknown	severe acute respiratory syndrome coronavirus 2 (SARS-CoV-2, COVID-19)
Principle symptoms	five major symptoms (Fissured lips/strawberry tongue, bilateral conjunctivitis, neck lymphadenopathy, limb induration, and skin rash)	dizziness, vomiting, abdominal pain, diarrhea, skin rash, cough, rhinorrhea, and conjunctival injection
Fever (>38°C)	100%	100%
Treatment	IVIG + aspirin(Steroids for high-risk group)	IVIG, steroids, anti-IL6, etc.
Age distribution	85% < 5 years old	9 years old (median)
Sex (gender)	Male > female	Male > female
Prevalence	Most common in Asia	Less common in Asia
Coronary aneurysm	3–5%	~14%

This table was modified and adapted from Chen et al. [73].

## Data Availability

Not applicable.

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
