# Peer review of "Diagnosis, Progress, and Treatment Update of Kawasaki Disease"

_ijms, 2023, doi:10.3390/ijms241813948_

Round 1

Reviewer 1 Report

In the Manuscript entitled "Diagnosis Progress and treatment update of Kawasaki Disease," the authors provide a review to discuss several important aspects related to Kawasaki Disease diagnosis and management. It's an exciting paper with a significant and relevant theme. However, the authors must observe some aspects before acceptance. In this way, the authors are encouraged to add an introduction that involves the work's principal elements and objectives of the work. 

Author Response

I have added the introduction and work's principal element.

The diagnosis and treatment of KD need update including diagnosis criteria comparison, the importance of eosinophil, novel markers and technique helping diagnosis, potential role of hydrogen gas inhalation in coronary artery lesions (CAL) and precise treatment to prevent CAL formation.

Reviewer 2 Report

While this paper is not a systematic review, it presents a comprehensive review of Kawasaki disease by experts in the field. The inclusion of the differences in diagnostic criteria between Japan and the U.S., as well as a unique mnemonic for the characteristic features of Kawasaki disease, make this work distinctive. The paper is generally well-written and requires minimal corrections. However, one major point of contention is Figure 1; the absence of descriptions for the x-axis and y-axis makes the interpretation of the figure unclear. It would be beneficial for the readers if the authors could provide clarity on this aspect. I would recommend revising this to enhance the paper's overall comprehensibility.

Author Response

I have added the x- and y-axis of figure 1. 

Reviewer 3 Report

The study reviews current knowledge regarding Kawasaki disease. The author has significant clinical experience in the field, supported by previous publications regarding KD. The study summarizes the clinical picture, diagnosis and treatment of the disease with additional tables and Figures.

I have several comments:

 - page 3 - abbreviation MIS-C - first time used, not explained

- Figure 1 - no legend and data, only colourfully bars are visible; needs correction

- Page 5 - I would use the phrase - fulfil criteria instead of satisfying them

In the end, the author presents the potential benefits of hydrogen in treating KD and MIS-C. It must be stressed that it is still an experimental method so far, which should be noted both in the text and in Table 4.

- Page 5 - I would use the phrase - fulfil criteria instead of satisfying them

Author Response

- page 3 - abbreviation MIS-C - first time used, not explained

--> we have added the explained of MIS-C.

- Figure 1 - no legend and data, only colourfully bars are visible; needs correction

--> We have revised the figure 1.

- Page 5 - I would use the phrase - fulfil criteria instead of satisfying them

--> We have revised it.

In the end, the author presents the potential benefits of hydrogen in treating KD and MIS-C. It must be stressed that it is still an experimental method so far, which should be noted both in the text and in Table 4.

--> We have revised it.

Reviewer 4 Report

A review of the diagnosis and treatment for Kawasaki Disease, described in detail. I would like to request a correction or addition to the following comment.

Major points

#1 Figure 1: It is not clear from the figure you gave us what symptoms were present and how frequent they were. The authors should illustrate it so that we can understand. In addition, the authors should include a citation in the illustration. Furthermore, do you have permission to do so?

#2 4 Echographic findings: The authors should provide a representative case or a diagram of the measurements, as this is the key to the diagnosis.

#3 BNP and its's-related markers are usually derived from the myocardium, and are usually elevated in inflammation, which may be a burden to the myocardium. Is there any evidence to suggest that these markers are direct markers of coronary artery disease?

#4 L450-455 lists a regimen of treatment at your facility, but was this decided based on the results of some study? Is this based on the results of some studies, or are you changing some treatment guidelines?

#5 Is the author the only one treating with hydrogen gas inhalation? The authors had better list any others that are being performed. However, if this is a case report from your institution only, the authors had better keep the description to the extent that there is a possibility of treatment effect.

Minor points

#1 L31. The authors should include references here.

#2 9. The subtitle of the potential role cites references (73). This is not common and should be corrected.

#3 The full spelling of Kawasaki Disease (KD), coronary artery lesions (CAL), intravenous immunoglobulin (IVIG), American Heart Association (AHA) has appeared many times, for example, L348, L373, L374, L376, L396, L414, L417, L421, L458, L459, etc. The authors should revise them. 

English in particular does not seem to be a problem. There are a few places where the same word is used in full spelling many times, which we have noted as noted above.

Author Response

Major points

#1 Figure 1: It is not clear from the figure you gave us what symptoms were present and how frequent they were. The authors should illustrate it so that we can understand. In addition, the authors should include a citation in the illustration. Furthermore, do you have permission to do so?

--> I have added the x- and y- axis of figure 1 that was made by myself (didn't need permission to use).

#2 4 Echographic findings: The authors should provide a representative case or a diagram of the measurements, as this is the key to the diagnosis.

--> we have added the figure of echographic findings.

#3 BNP and its's-related markers are usually derived from the myocardium, and are usually elevated in inflammation, which may be a burden to the myocardium. Is there any evidence to suggest that these markers are direct markers of coronary artery disease?

--> As mention in the manuscript: It is recognized as a well-established biomarker for both congestive heart failure and coronary artery diseases.

#4 L450-455 lists a regimen of treatment at your facility, but was this decided based on the results of some study? Is this based on the results of some studies, or are you changing some treatment guidelines?

--> As mentioned in the manuscript: Administering a second dose of IVIG (2 g/kg over 10-12 hours, following the same dosage and treatment duration as the initial IVIG) is a consideration (65, 66), methylprednisolone pulse therapy (67), tumor necrosis factor-alpha blockade (68), cyclophosphamide, cyclosporine A, plasmapheresis (69), methotrexate (70), and plasma exchange (71) have all been reported to be beneficial for KD patients with initial IVIG-resistance. (these regiment were based on results of study as citation reference)

IVIG is recognized as the primary standard treatment for KD in accordance with AHA and Japanese guidelines.

However, there is currently no established standard treatment guideline for KD cases resistant to IVIG. (We didn't change current guideline)

#5 Is the author the only one treating with hydrogen gas inhalation? The authors had better list any others that are being performed. However, if this is a case report from your institution only, the authors had better keep the description to the extent that there is a possibility of treatment effect.

--> Hydrogen's antioxidant and anti-inflammatory effects might contribute to cardiovascular protection and as a possibility of treatment effect for KD. The potential therapeutic impact of hydrogen gas inhalation in patients with KD, whether with or without CAL formation, also demands further investigation.

Minor points

#1 L31. The authors should include references here.

--> we have added ref. (2)

#2 9. The subtitle of the potential role cites references (73). This is not common and should be corrected.

--> We have moved the ref (73).

#3 The full spelling of Kawasaki Disease (KD), coronary artery lesions (CAL), intravenous immunoglobulin (IVIG), American Heart Association (AHA) has appeared many times, for example, L348, L373, L374, L376, L396, L414, L417, L421, L458, L459, etc. The authors should revise them.

--> we have revised it.
